# Single cell heterogeneity in influenza A virus gene expression shapes the innate antiviral response to infection

Jiayi Sun[1☯], J. Cristobal Vera[1,2☯], Jenny Drnevich[3], Yen Ting Lin[4], Ruian Ke[5], Christopher B. Brooke[1,2]*

**1** Department of Microbiology, University of Illinois at Urbana-Champaign, Urbana, Illinois, United States of America, **2** Carl R. Woese Institute for Genomic Biology, University of Illinois at Urbana-Champaign, Urbana, Illinois, United States of America, **3** High-Performance Biological Computing at the Roy J. Carver Biotechnology Center, University of Illinois at Urbana-Champaign, Urbana, Illinois, United States of America, **4** Information Sciences Group, Computer, Computational and Statistical Sciences DIvision (CCS-3), Los Alamos National Laboratory, Los Alamos, New Mexico, United States of America, **5** T-6, Theoretical Biology and Biophysics, Los Alamos National Laboratory, Los Alamos, New Mexico, United States of America

☯ These authors contributed equally to this work.
* cbrooke@illinois.edu

**Data Availability Statement:** All next generation sequencing data is available from the NCBI GEO via accession number GSE143167. All other relevant

## Abstract

Viral infection outcomes are governed by the complex and dynamic interplay between the infecting virus population and the host response. It is increasingly clear that both viral and host cell populations are highly heterogeneous, but little is known about how this heterogeneity influences infection dynamics or viral pathogenicity. To dissect the interactions between influenza A virus (IAV) and host cell heterogeneity, we examined the combined host and viral transcriptomes of thousands of individual cells, each infected with a single IAV virion. We observed complex patterns of viral gene expression and the existence of multiple distinct host transcriptional responses to infection at the single cell level. We show that human H1N1 and H3N2 strains differ significantly in patterns of both viral and host anti-viral gene transcriptional heterogeneity at the single cell level. Our analyses also reveal that semi-infectious particles that fail to express the viral NS can play a dominant role in triggering the innate anti-viral response to infection. Altogether, these data reveal how patterns of viral population heterogeneity can serve as a major determinant of antiviral gene activation.

## Author summary

The combination of the enormous diversity of viral populations, underlying heterogeneity within host cells, and random chance mean that no two infected cells look exactly the same. The role that single cell heterogeneity plays in determining infection outcomes remains very poorly understood. Here, we describe an approach that allowed us to quantify both viral and host gene expression in thousands of influenza A virus infected cells. We observed an enormous degree of heterogeneity in both viral and host gene expression between individual infected cells. We found that virions that fail to express the viral NS gene can be major drivers of host antiviral immune activation. Comparison of antiviral

data are within the manuscript and its Supporting Information files.

**Funding:** This work was partially supported by a National Institute of Allergy and Infectious Disease grant 1R01AI139246-01A1 (C.B.B.), the Defense Advanced Research Projects Agency INTERCEPT program through contracts R-00676-19-0 (R.K.) and W911NF-17-2-0034 (C.B.B.), and a gift from the Roy J. Carver Charitable Trust 17-4905 (C.B. B.). The funders had no role in study design, data collection and analysis, decision to publish, or preparation of the manuscript.

**Competing interests:** The authors have declared that no competing interests exist.

gene expression patterns between H1N1 and H3N2 infection revealed surprising differences in single cell patterns of innate immune activation. Altogether, these studies identify patterns of single cell heterogeneity in both host and viral gene expression as a significant driver of infection outcomes.

## Introduction

RNA virus populations typically contain an enormous amount of sequence diversity due to an absence of virally encoded proofreading activity [1]. In some cases, this genetic diversity can significantly influence infection outcomes [2–4]. In addition, influenza A viruses (IAV) also exhibit substantial heterogeneity in the gene expression patterns of individual virions [5]. Most IAV virions are only capable of expressing variable, incomplete subsets of viral genes [6]. The production and gene expression patterns of these semi-infectious particles (SIPs) can vary significantly between IAV strains [7]. Altogether, the high degree of diversity within IAV populations means that patterns of viral gene expression can vary significantly between individual infected cells.

Numerous studies have leveraged recent advances in single cell analysis methods to assess the extent of cellular heterogeneity present during infection by a variety of viruses, including IAV [8–17]. These studies connect to a larger body of work that has begun to explore single cell heterogeneity within different host cell populations and tissues [18–20]. The extent to which IAV population diversity influences patterns of single cell heterogeneity and the ways in which these patterns shape broader infection dynamics and outcomes remain poorly understood.

Here, we query the combined viral and host transcriptomes from thousands of individual cells, each infected by a single virion. This high-resolution dissection of viral and host gene expression patterns reveals that the transcriptional responses of individual infected cells can be highly divergent, resulting from the interplay between underlying cellular heterogeneity and viral population diversity. Thus, the host response to IAV infection consists of a heterogenous assemblage of highly variable single cell responses. This approach also reveals critical differences in the interactions of H1N1 and H3N2 viruses with the host antiviral machinery at the single cell level and implicates heterogeneity in NS segment expression as a major driver of the innate antiviral response to IAV infection. Altogether, these results establish the interplay between viral and host cell heterogeneity as a critical determinant of cellular infection outcomes.

## Results

### Generation of viral and host transcriptional data from thousands of singly infected cells

To assess the effects of viral population heterogeneity on the host response to infection, we examined the combined viral and host transcriptional profiles from thousands of single infected cells. To focus on the effects of viral heterogeneity, we wanted to remove the variability that could arise from variation in cellular MOI [21]. To ensure that the vast majority of infected cells were each infected with a single virion, we infected A549 cells with either the 2009 H1N1 pandemic strain A/California/07/2009 (Cal07), or the seasonal human H3N2 strain A/Perth/16/2009 (Perth09) at an MOI of 0.01, and blocked secondary spread in the culture through the addition of $NH_4Cl$ [22]. This resulted in only a tiny fraction of cells being

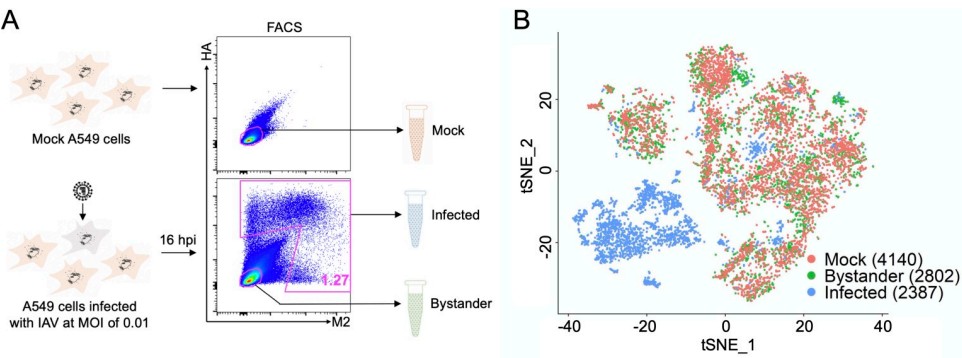

**Fig 1. Generation of viral and host transcriptional data for thousands of singly infected cells. (A)** Schematic depicting our strategy for generating single cell RNAseq libraries from thousands of cells infected at low MOI. In brief, we infect A549 cells with Cal07 or Perth09 at MOI = 0.01 to ensure that infected cells are infected with a single virion. We then block secondary spread with $NH_4Cl$ treatment to make sure infection timing is uniform across all infected cells. Finally, we sort "infected" and "bystander" cells based on surface expression of HA and/or M2 and immediately generate single cell RNAseq libraries from these sorted cell populations using the 10X Chromium device. In parallel, mock cells are sorted and used as uninfected controls. **(B)** tSNE dimensionality reduction plot showing the extent of overlap between 3 indicated cell populations from Perth09 experiment clustered based on transcriptional similarity.

infected, so to obtain sufficient numbers of infected cells for our analysis, we enriched infected cells by sorting based on surface expression of HA and/or M2 (**Fig 1A**). This approach had the added benefit of allowing us to sort HA-M2- bystander cells from the same culture to assess the paracrine effects of infection. In parallel, we sorted a group of mock-infected cells as a control.

We used the 10X Genomics Chromium platform to generate oligo-dT-primed single cell RNAseq libraries for mock, infected (HA+ and/or M2+), and bystander (HA-M2-) cells and sequenced them on an Illumina Novaseq. We demultiplexed the reads and mapped them to a customized hybrid reference containing both human and influenza sequences/annotation. Following quality control filtering, we had high quality combined viral and host transcriptomes from thousands of single virion-infected cells, as well as uninfected bystander and mock cells. Clustering of these three libraries by transcriptional similarity revealed that infected and uninfected cells largely clustered independently, as would be expected (**Fig 1B, S1 Fig**). For Cal07 (but not Perth09), mock and bystander cells clustered independently, but it is possible this is due to a batch effects, as the mock and bystander cells came from different culture vessels.

Although we sorted infected cells based on the presence of viral markers, it is possible that some uninfected cells were included within the infected cell libraries due to a combination of cell sorting errors, library index hopping, or cross-contamination with mRNA released from dying cells. To identify the truly-infected cells within the infected libraries, we examined the distributions of percentage of viral counts within each cell (S2 **Fig**). For cells in infected libraries of both Cal07 and Perth09, we observed clear bimodal distributions in the percentage of total viral counts. We assumed that cells within the lower peak (percentages of viral reads: < 1% for Cal07 and < 3% for Perth09) were actually uninfected. We calculated kernel density estimates on these distributions and used the first local minima to generate cutoff thresholds to differentiate truly-infected cells from uninfected cells (**S2A and S2C Fig**). Cells from infected libraries that fell below these thresholds clustered with the bystander cells, suggesting that these cells were truly uninfected (**S2B and S2D Fig**). We used this approach in subsequent analyses to define truly-infected cells; however, it is possible that a small fraction of the cells we

call as uninfected by these criterion are also truly infected yet exhibit very low levels of viral gene expression.

## Enormous single cell heterogeneity in viral gene expression patterns

We first asked whether we observed the same degree of heterogeneity in viral gene expression between infected cells that has been reported previously [6,10,12]. We calculated the fraction of all transcripts within each cell that were viral in origin (**Fig 2A**). Similar to prior studies, total viral transcript levels ranged enormously between individual cells: from <1% up to ~90%. Notably, this heterogeneity arose under conditions where both viral input (1 virion/cell) and infection timing were largely equivalent across all cells.

Interestingly, the distributions for Cal07 and Perth09 were quite different, as Perth09-infected cells tended to have a much higher fraction of viral transcript. This could be a reflection of differences in gene expression kinetics, or of differences in the relative capacities of the two viruses to exploit the host cell machinery necessary for viral gene expression. This pattern of extreme cell-to-cell heterogeneity was also observed for individual viral transcripts, again consistent with previous reports (**Fig 2B and 2C**)[12]. We examined the degree to which individual pairs of viral genes were correlated with each other within infected cells that expressed all viral genes (**S3** and **S4** **Figs**). For Cal07, we found that most pairs of viral transcripts were fairly

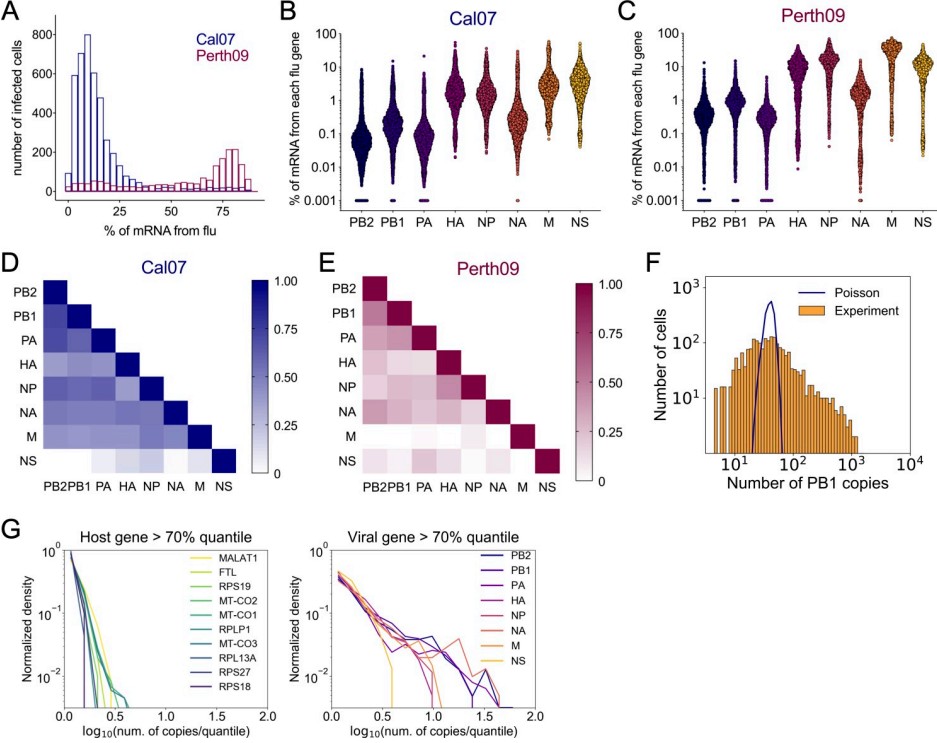

**Fig 2. Enormous heterogeneity in viral gene expression patterns. (A)** Distributions of Cal07 and Perth09-infected A549 cells, binned by the fraction of total cellular poly(A) RNA that is viral in origin. **(B)** Plots show the fraction of total poly(A) RNA per cell that maps to the indicated viral gene segment of Cal07. Each dot represents a single cell, cells with no detectable reads mapping to the indicated segment were arbitrarily assigned a value of 0.001 to show up on the log10 scale. **(C)** Same figure as (B) for Perth09. **(D)** $R^2$ correlation values plotted as heat map for all pairwise comparisons of Cal07 viral transcripts within infected cells positive for all viral genes. **(E)** Same figure as (D) for Perth09. **(F)** Distribution of normalized Cal07-PB1-derived reads per cell (orange) compared with a Poisson distribution of equal mean (blue line) on a log-log scale. **(G)** Distributions of top 10 most abundant host transcripts (left panel) and Cal07 viral gene expression (right panel) normalized by the 70[th] quantile on a log-log scale.

well correlated, with one obvious exception: the NS segment (**Fig 2D**). Expression levels of the individual viral genes were generally less well correlated during Perth09 infection, compared with Cal07, and expression of both the M and NS segments by Perth09 was especially poorly correlated with the other viral genes (**Fig 2E**).

We also found that the expression distributions for individual viral genes were highly over-dispersed compared with a Poisson distribution (**Fig 2F**). The tails of the expression distributions for several viral genes were an order of magnitude longer than those of the top 10 most abundant host transcripts (**Fig 2G**), emphasizing that viral gene expression is substantially noisier than host gene expression.

## Significant heterogeneity in the host transcriptional response to infection

We next asked whether the observed variation in viral gene expression levels was associated with variation in host cell transcription. To test this, we first clustered all infected cells based on their host transcription patterns. We identified multiple distinct transcriptional response groups to infection by either Cal07 or Perth09, demonstrating that there is not one standard transcriptional response to IAV infection and that a single cell type can simultaneously generate multiple distinct responses to the same virus population (**Fig 3A and 3D**). It should be noted that these cluster definitions are not absolute and that substantial heterogeneity in the normalized expression levels of individual host genes existed within individual clusters.

We next asked whether the cell clustering patterns were correlated with levels of viral gene expression and/or cell cycle status. For Cal07, we did observe a relationship between overall viral gene expression levels and cluster structure, as the majority of cells with high levels of viral gene expression were concentrated in cluster 5 (**Fig 3B**). The clustering patterns of Cal07-infected cells also appeared to be partially influenced by cell cycle status, with cells in G2M phase disproportionately falling into cluster 0 (**Fig 3C**). While it is not surprising that cell cycle status would contribute to transcriptional heterogeneity during infection, these data highlight how little is known about how cell cycle status may influence the cellular response to infection. In contrast, we did not observe any clear relationships between viral gene expression levels, cell cycle status, and clustering pattern for Perth09-infected cells (**Fig 3E and 3F**).

We also observed the existence of multiple distinct clusters within mock cells (**Fig 3G**), raising the question of whether the observed heterogeneity in infected cells was simply a reflection of the intrinsic heterogeneity of the cell population prior to infection. In other words, does infection actually increase the overall heterogeneity in host gene expression patterns beyond that seen in mock cells? To quantify and compare heterogeneity in overall host transcription between the two cell populations, we calculated the multivariate homogeneity of dispersions for mock and infected cells using their host gene expression profiles (**Fig 3H**)[23]. We found that infection significantly increased the overall single cell heterogeneity in host cell transcription patterns ($p < 10^{-15}$ by pairwise t test). Altogether, our data are consistent with a model in which the interaction between viral population heterogeneity and pre-existing host cell heterogeneity gives rise to multiple distinct transcriptional responses at the single cell level.

## Substantial heterogeneity in expression patterns of critical determinants of IAV infection outcome, including IFNs and ISGs

To explore the specific transcriptional differences that distinguished different clusters of infected cells, we performed single cell differential gene expression analysis which allowed us to compare the expression of host transcripts between different clusters [24,25]. This approach revealed that the induction of numerous host genes known or likely to be involved in shaping IAV infection outcome varied significantly between clusters (**Fig 4A and 4C**). Most strikingly,

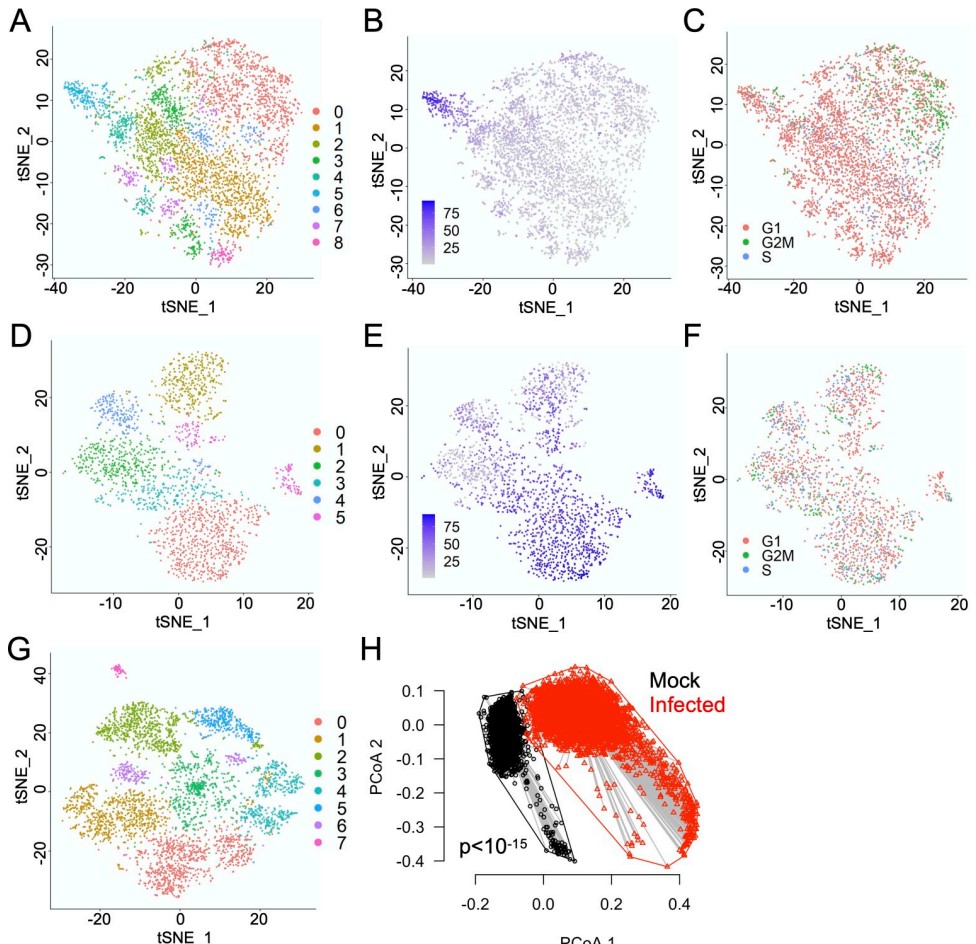

**Fig 3. Significant heterogeneity in the host transcriptional response to infection. (A)** tSNE dimensionality reduction plot showing all Cal07-infected A549 cells clustered based on similarity of host transcription patterns. **(B)** Same tSNE plot of Cal07-infected cells shown in (A) with each cell colored by the percentage of total cellular poly(A) RNA that is viral in origin. **(C)** Same tSNE plot of Cal07-infected cells shown in (A) with each cell colored by predicted cell cycle stage, as determined by the Scran package. **(D)** Same figure as (A) for Perth09. **(E)** Same figure as (B) for Perth09. **(F)** Same figure as (C) for Perth09. **(G)** tSNE dimensionality reduction plot showing mock A549 cells from Cal07 experiment clustered based on similarity of host transcriptional patterns. **(H)** Principle coordinate axes (PCoA) plot comparing the multivariate dispersions for mock (black) and Cal07-infected (red) A549 cells. The first two axes (PCoA 1 and PCoA 2) in the multivariate homogeneity of group dispersions analysis are used.

for Perth09 (but not Cal07), expression of type III IFNs and several ISGs were heavily concentrated within a single cluster, cluster 5 (**Fig 4C**). Beyond obvious factors like the IFNs and ISGS, both viruses, exhibited highly heterogeneous expression patterns of numerous other host genes likely to influence infection outcomes. For example, per-cell expression levels of NEAT1, a long non-coding RNA (lncRNA) involved in inflammasome formation, regulation of cytokine and chemokine expression, and nuclear paraspeckle formation [26–31], varied significantly between infected cell clusters for both viruses (**Fig 4B and 4D**).

Hundreds of other host genes exhibited similarly heterogeneous patterns of expression between individual infected cells (**S1 Table, S2 Table**), though it must be pointed out that these measurements can be skewed somewhat by variation in viral RNA levels. Altogether, these data clearly demonstrate how numerous drivers of infection outcomes (not just IFNs) may be primarily expressed by limited subsets of infected cells.

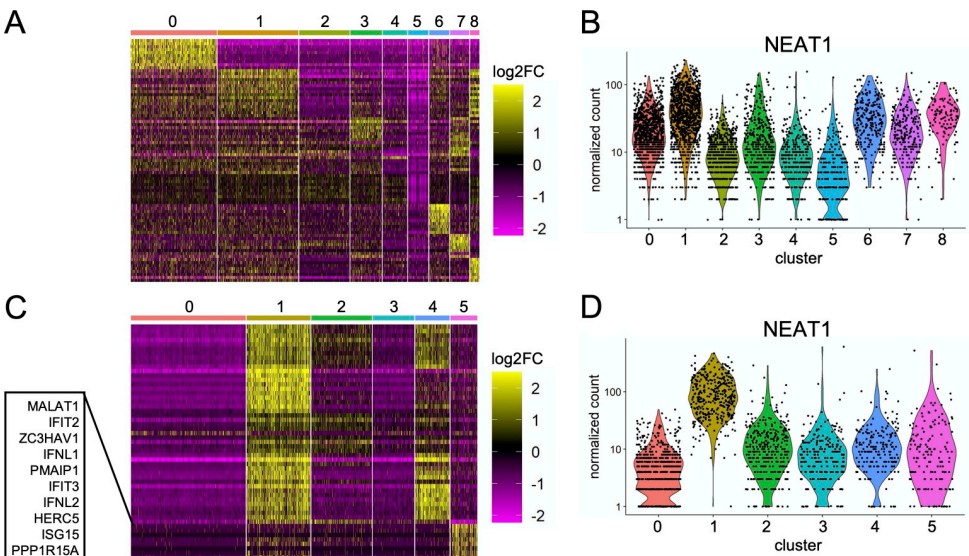

**Fig 4. Substantial virus strain-specific variation in the expression patterns of critical determinants of IAV infection outcome. (A)** Heat map showing differential expression of the top 10 characteristic host genes for each cluster of Cal07-infected cells (from Fig 3A). Individual cells are each represented by a column, grouped by cluster, with individual rows representing relative expression of the top 10 specific host transcripts most significantly (lowest p values) associated with each cluster. **(B)** Comparison of normalized per cell counts of NEAT1 between clusters of Cal07-infected cells shown in (A). **(C)** Same heat map as (A) for each cluster from Perth09-infected cells (From Fig 3D), with the top 10 genes defining cluster 5 highlighted. **(D)** Comparison of normalized per cell counts of NEAT1 between clusters of Perth09-infected cells shown in (C).

## Many infected cells have undetectable levels of one or more viral transcripts

The vast majority of IAV virions fail to express one or more viral genes, resulting in the expression of variable, incomplete subsets of viral gene products under low MOI conditions [5–7,32]. We asked whether this variation in functional viral gene content within individual cells contributes to the observed heterogeneity in host gene transcription. To assess the presence or absence of each individual viral gene within infected cells while avoiding false positives due to RNA cross-contamination, we used the same approach that we used to determine infection status of cells in infected libraries. Just as with overall viral gene expression, nearly all individual viral genes exhibited clear bimodal patterns in the distributions of viral gene percentage within each cell on a log scale, suggestive of clear separations between positive and negative populations and allowing us to set cutoff thresholds based on the first local minima (S5A and S5B Fig). The only exceptions were PB2 and PA of Cal07. For these two genes that did not exhibit clear bimodal distribution, we set the cutoff thresholds at -1.5 (log10 scale) based on the distributions of polymerase gene expression in Perth09 and uninfected cells within the infected library of Cal07. It must be noted that while these cutoff thresholds represent our best effort to define the expression statuses of individual viral genes, they may not be perfectly accurate.

Using these cutoffs, we found that the fractions of infected cells that failed to express detectable transcripts ranged from ~35% to ~5% for the individual viral gene segments (**Fig 5A and 5C**). It is highly likely that our method of enriching for infected cells by sorting based on high level HA and/or M2 expression significantly biased these results. Regardless, we still observed substantial numbers of infected cells that fail to express detectable levels of individual viral transcripts, with roughly half of all infected cells failing to express detectable levels of at least

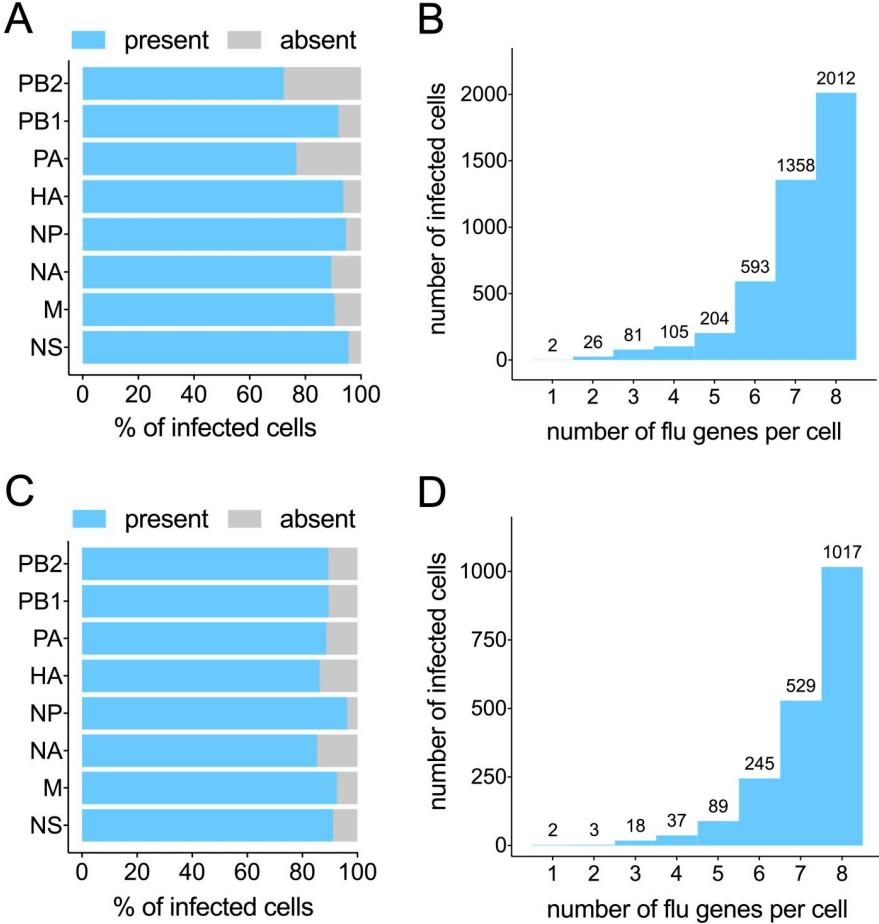

**Fig 5. Many infected cells have undetectable levels of one or more viral transcripts. (A)** The percentage of all Cal07-infected cells called as positive for the indicated gene segments **(B)** All Cal07-infected cells binned by the total number of viral gene segments that were called positive, with the actual numbers of cells in each group detailed above. **(C)** Same figure as (A) for Perth09. **(D)** Same figure as (B) for Perth09.

one viral gene (**Fig 5B and 5D**). We were surprised to see so many cells that lacked detectable levels of the polymerase transcripts because we expected that sorting infected cells based on surface protein expression would bias cell collection against cells that could not synthesize new polymerase complexes. One important caveat with our approach is that we assessed transcript levels at 16 hpi and thus could have missed transient early expression of viral genes.

## The absence of individual viral genes has a significant effect on the host transcriptional response to infection

We next asked how the expression status of individual viral genes influences the overall host response to infection to both Cal07 and Perth09. We grouped all infected cells into positive and negative populations based on the expression of each individual viral gene segment and compared host transcript expression between the two infected cell populations. For each viral gene, we generated a list of host transcripts that were reported as significantly different between positive and negative cells by both NBID and MAST tests (**S3–S16 Tables**). This approach allowed us to tease out the effects of individual viral gene segments from the more general effects of infection.

We identified hundreds of host genes with expression levels that varied significantly based on the expression status of individual viral genes (**S6A and S6B Fig**). Closer examination revealed that many of these hits were found for multiple viral genes, suggesting that they may correlate with overall viral gene expression levels or co-expression of multiple viral gene segments. When we focused on the host genes that only exhibited differential expression in association with the expression status of a single viral gene segment, we found that most were associated with the PA and NS segments (**Fig 6A and 6D**).

## NS segment expression status is a major determinant of IFN and ISG induction by Perth09, but not Cal07

We focused on the NS segment, as the NS-encoded NS1 protein plays a well-established, multi-functional role in manipulating the host cell environment and the anti-viral response and could thus serve as a positive control for our approach [33–36]. We found that approximately 4% of Cal07-infected cells and 9% of Perth09-infected cells had background or undetectable levels of NS segment transcript, and most of them came from cluster 5 with high viral transcript level which was in consistent with low correlation value between NS and other segments (**Fig 6B and 6E**). For Cal07, we observed over 300 host genes whose expression was uniquely influenced by NS. Notably, the expression of specific IFNs or ISGs was not significantly affected by NS expression status during Cal07 infection. In contrast, we did observe a significant reduction in SLFN5 expression frequency in infected cells that failed to express NS compared with those that did express NS (**Fig 6C**). SLFN5 is an interferon-stimulated gene (ISG) shown to negatively regulate STAT1-dependent anti-viral gene transcription [37]. These results both validate the utility of our approach and identify the suppressor of antiviral gene transcription SLFN5 as a novel host target of the NS gene segment during Cal07 infection.

In contrast with Cal07 infection, NS segment expression status for Perth09 was significantly correlated with expression levels of numerous IFNs, ISGs, and other innate immune factors (**Fig 6F**). For example, the type III IFNs IFNL1 and IFNL2 were expressed by ~20% of Perth09-infected cells lacking detectable NS expression, but both were only observed in ~2% of infected cells that did express NS. We also observed similar NS-dependent expression patterns for several ISGs (e.g. IFIT1, IFIT2, IFIT3, ISG15, ZC3HAV1, and OAS1), as well as the neutrophil-recruiting chemokine CXCL1. Thus, for Perth09 but not Cal07, single cell heterogeneity in NS segment expression status is a major determinant of the activation of the innate antiviral response.

## H1N1 and H3N2 strains can differ significantly in single cell patterns of IFN and ISG transcription

Finally, we more broadly compared the activation of host innate anti-viral gene expression at the single cell level between Cal07 and Perth09. In Cal07-infected cells, expression of both type I and III IFNs were extremely rare, and only IFNL1 passed our quality control filter (detected in more than 4 cells) (**Fig 7A**) [13,38]. For type I IFN (but probably not type III IFN), this could partially be a function of the relatively late timepoint that we examined, as IFNβ expression typically peaks earlier during infection [39]. Similarly, expression of multiple ISGs was also very rare in Cal07-infected cells, indicating a near-complete failure to initiate an innate anti-viral response. In bystander cells, both IFN and ISG expressions were also minimal, suggesting that the inhibition of IFN induction by Cal07 was sufficient to largely prevent paracrine ISG activation (**Fig 7B**).

In Perth09-infected cells, IFNL1 (type III IFN) transcription was roughly 20-fold more frequent compared with Cal07, indicating a significant difference in the ability of the host to

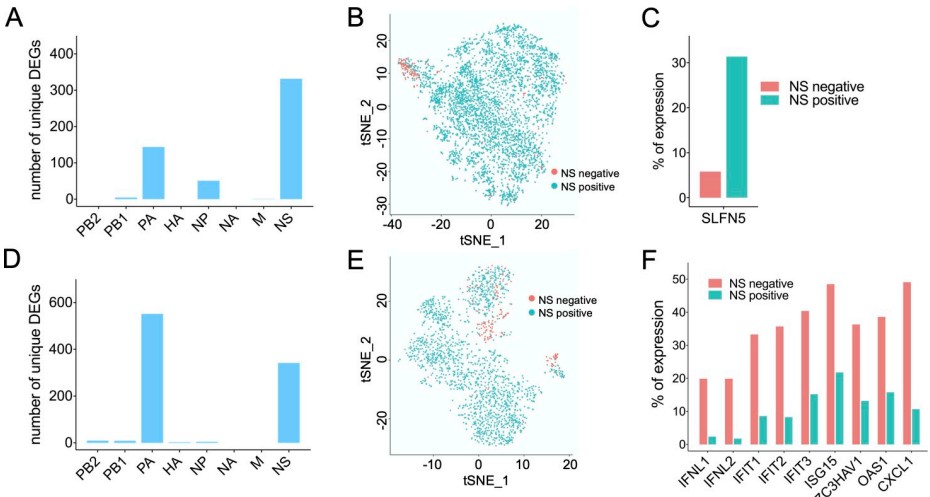

**Fig 6. Dissection of the effects of individual viral gene expression on the host transcriptional response to infection.** **(A)** The number of host transcripts for which expression levels significantly differ depending on whether the indicated Cal07 gene segment is present or not, according to both MAST and NBID (host genes that are differentially regulated by the expression status of more than one viral segment are excluded). **(B)** tSNE plot of all Cal07-infected A549 cells colored based on whether NS segment-derived transcripts are detected (Cyan) or not detected (Salmon). **(C)** Percentages of NS negative and NS positive Cal07-infected A549 cells that have detectable levels of SLFN5. **(D)** Same figure as (A) for Perth09. **(E)** Same figure as (B) for Perth09. **(F)** Percentages of NS negative and NS positive Perth09-infected A549 cells that have detectable levels of the indicated host transcripts.

initiate an IFN response to infection by these two strains (**Fig 7C**). Consistent with this, some (but not all) ISGs were clearly expressed more frequently in bystander and Perth09-infected cells compared with Cal07, suggesting that Perth09 is less able to prevent paracrine activation

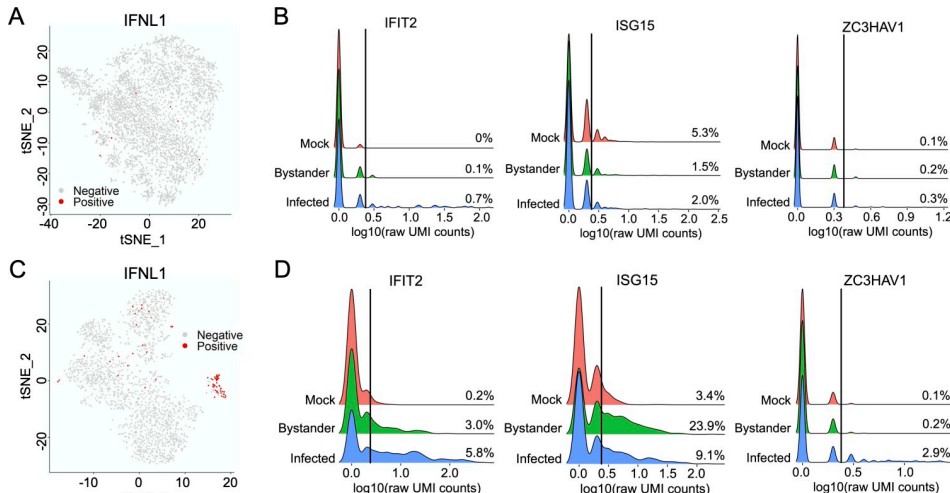

**Fig 7. H1N1 and H3N2 strains differ significantly in single cell patterns of IFN and ISG transcription.** **(A)** tSNE plots of all Cal07-infected A549 cells with cells that have detectable level of IFNL1 colored in red. **(B)** Histograms comparing distributions of per-cell UMI counts in log10 scale for the indicated host transcripts across the three libraries (mock, bystander, and infected) for Cal07 experiment. Cells with zero count excluded to avoid dominating the y-scale. Vertical lines indicate cutoff thresholds of 3 UMI counts, with percentages of all cells (including cells with zero count) in each library above the threshold shown on right. **(C)** Same figure as (A) for Perth09. **(D)** Same figure as (B) for Perth09.

(Fig 7D). Altogether, these data reveal clear differences in single cell patterns of IFN and ISG transcription between the human H1N1 and H3N2 strains tested.

## Discussion

The host response to infection arises from the combined responses of many individual cells, both infected and uninfected. To understand the factors that govern host responses at the tissue and organismal levels, it is critical to define patterns of variation in single cell infection responses. Here, we demonstrate that IAV infection gives rise to a heterogeneous collection of divergent transcriptional responses at the single cell level. Notably, this heterogeneity occurred within a single cell type, under conditions where per-cell viral input and infection timing were normalized. Single cell responses *in vivo* may be significantly more variable. By comparing patterns of viral gene expression at the single cell level between two distinct strains of human IAV, we demonstrate how viral population heterogeneity can be a major driver of innate immune activation, in a virus strain-specific manner. These data establish a clear role for viral heterogeneity in modulating the host response to infection and highlight the power of single cell approaches to reveal new determinants of viral infection outcomes.

Variation in gene expression patterns between individual infected cells appears to arise from a combination of (1) genetic and genomic variation between individual virions, (2) underlying heterogeneity within host cells, and (3) stochastic variation in infection processes. In line with previous reports of significant genetic and phenotypic heterogeneity within A459 and other transformed cell line populations, we observed substantial transcriptional heterogeneity between uninfected A549 cells that could not be simply explained by cell cycle status [40–42]. As a result, virions that enter different cells establish infection within differing cellular environments that may be more or less supportive and primed for differing transcriptional responses to infection. This effect is likely enhanced by the stochastic variation inherent in different viral life cycle stages [43].

Layered on top of this intrinsic cellular heterogeneity and stochasticity is the genetic and genomic heterogeneity characteristic of IAV populations [5]. We and others have previously demonstrated that well-characterized viral modulators of host cell function such as NS1 and PA-X are not ubiquitously expressed during infection [6,12]. Here, we show that this variability in viral gene expression between individual infected cells has clear consequences for the host cell transcriptional response. It is clear that future efforts to understand the role of critical viral proteins such as NS1 in shaping infection outcome will have to account for heterogeneity in single cell expression patterns.

The comparison of Cal07 and Perth09 revealed clear commonalities and differences in single cell expression patterns between the two viruses that may have relevance for understanding differences in the apparent pathogenicity of different human H1N1 and H3N2 viruses. The most obvious difference was seen in the activation of IFN and ISG transcription. In addition, while NS segment expression status significantly influenced the host response to both viruses, only Perth09 showed a clear relationship between NS levels and the activation of the host antiviral machinery. This finding is in line with previous studies that showed that (a) different human IAV strains differ in bulk induction of IFNβ, and (b) the 2009 pandemic H1N1 NS1 protein was less effective at suppressing transcription of human IFNs and ISGs compared with NS1 genes from other human IAV isolates [44,45]. While *de novo* NS segment transcription appeared to be largely dispensable for blocking IFN and ISG upregulation by Cal07 (at least under the conditions examined in this study), overall expression frequencies for select IFNs and ISGs were actually lower during Cal07 infection compared with Perth09, indicating that Cal07 is more effective at suppressing or evading IFN activation in A549 cells. Yet, it remains

an open question how overall viral gene expression level might influence the host innate antiviral response, given the clear differences we observed between the two viruses. Overall, our data make clear that there is still a lot that we do not understand about the interplay between viral gene expression patterns, viral genotype, and the host antiviral response.

In this study, we took steps to eliminate two other sources of viral heterogeneity that are likely common during natural IAV infection. The first is presence of defective interfering particles (DIPs), which we minimized by generating our virus stocks under low MOI conditions, and subsequently verified (**S7 Fig**). DIPs appear to be common within IAV populations, even in humans, and can have complicated effects on both viral and host gene expression [13,46–51]. We would predict that the presence of DIPs would further increase overall heterogeneity between individual cell responses to the virus. It is worth noting that we could not reliably detect the presence of DIP-associated transcripts within our data due to the unavoidably incomplete coverage of the viral gene segments generated by our short-read 3'-enriched sequencing approach. The second likely source of additional heterogeneity that we excluded from this study is variability in the cellular MOI. We and others have demonstrated that cellular co-infection can be common *in vivo* [7,52–54]. This suggests that the number of viral genomes entering individual cells is likely quite variable. We recently showed that this variability in cellular MOI can have distinct phenotypic consequences, both for viral replication dynamics and for IFN induction [21]. Altogether, it appears likely that the heterogeneity that we describe here underrepresents what would be observed *in vivo*.

Our results extend our previous results and those of other groups in establishing the enormous amount of heterogeneity in viral gene expression that occurs at the single cell level, even under experimental conditions designed to minimize sources of variability [6,10,12,43,55]. Critical phenotypes such as viral load dynamics, transmissibility, and pathogenicity must emerge from the collective output of heterogeneous populations of infected cells. This raises questions of how selection may act upon patterns of viral heterogeneity to alter these emergent phenotypes and the extent to which these heterogeneity patterns are under viral genetic control. These questions are especially pertinent for segmented viruses like IAV but are relevant across diverse virus families.

Altogether, our results help establish the importance of considering the roles of viral and host cell heterogeneity in influencing the pathogenesis of viral infections. Similar to the way that viral populations are now viewed, our data clearly establishes that the host response to infection should be seen as a heterogeneous assemblage of single cell responses that collectively give rise to the bulk phenotypes that are generally measured. This creates the potential for complex interactions between responding cell subsets and raises the question of how such a heterogeneous system is effectively regulated. It also raises the possibility that viral and host response dynamics may be disproportionately driven by small subsets of cells that are obscured during bulk analysis. Dissection of these diverse constituents is likely to reveal new mechanisms that govern the pathogenesis of influenza virus infection.

## Methods

### Plasmids

The A/California/04/09 and A/Perth/16/2009 reverse genetics plasmids were generous gifts from Drs. Jonathan Yewdell and Seema Lakdawala, respectively. Plasmids encoding A/California/07/09 were generated by introducing A660G and A335G substitutions into HA and NP respectively, to match the amino acid sequences of A/California/07/09 HA and NP (NCBI accession numbers CY121680 and CY121683).

## Cells

Madin-Darby canine kidney (MDCK) and human embryonic kidney HEK293T (293T) cells were maintained in Gibco's minimal essential medium with GlutaMax (Life Technologies). Human lung epithelial A549 cells were maintained in Gibco's F-12 medium (Life Technologies). MDCK and A549 cells were obtained from Dr. Jonathan Yewdell; 293T cells were obtained from Dr. Joanna Shisler. All media were supplemented with 8.3% fetal bovine serum (Seradigm). Cells were grown at 37˚C and 5% $CO2$.

## Viruses

Recombinant A/California/07/09 (Cal07) and A/Perth/16/2009 (Perth09) viruses were rescued via the 8-plasmid reverse genetics approach. For the rescue of both viruses, sub-confluent 293T cells were co-transfected with 500 ng of the following plasmids: pDZ::PB2, pDZ::PB1, pDZ::PA, pDZ::HA, pDZ::NP, pDZ::NA, pDZ::M, and pDZ::NS, using JetPrime (Polyplus) according to the manufacturer's instructions. Plaque isolates derived from rescue supernatants were amplified into seed stocks in MDCK cells. Working stocks were generated by infecting MDCK cells at an MOI of 0.0001 $TCID_{50}$/cell with seed stock and collecting and clarifying supernatants at 48 hpi. All viral growth was carried out in MEM with 1 μg/ml trypsin treated with L-(tosylamido-2-phenyl) ethyl chloromethyl ketone (TPCK-treated trypsin; Worthington), 1 mM HEPES, and 100 μg/ml gentamicin. The titers of the virus stocks were determined via standard 50% tissue culture infectious dose ($TCID_{50}$) assay.

## Viral infection and cell sorting for single cell RNAseq

Confluent A549 cell monolayers in 3 T-25 flasks were infected with Cal07 (or Perth09) at MOI of 0.01 $TCID_{50}$/cell for 1 h. At 1 hpi, monolayers were washed with phosphate-buffered saline (PBS) and incubated in serum-containing F-12 medium. At 3 hpi, the medium was changed to MEM with 50 mM HEPES and 20 mM $NH_4Cl$ to block viral spread. At 16 hpi, monolayers were trypsinized and combined into single-cell suspension and washed with PBS. Cal07-infected cells were stained with Alexa Fluor 488-conjugated mouse anti-HA monoclonal antibody (mAb) EM4-CO4 (gift from Dr. Patrick Wilson) and Alexa Fluor 647-conjugated mouse anti-M2 mAb O19 (gift from Dr. Jonathan Yewdell). Perth09-infected cells were first stained with human anti-HA stem antibody FI6 (Gift from Dr. Adrian McDermott) and then stained with Alexa Fluor 488-conjugated donkey anti-human IgG (Jackson ImmunoResearch). After staining, cells were washed with PBS twice, and single live cells were sorted as "infected" or "bystander" populations based on the expression of HA and M2 on a BD FACS ARIA II sorter. Importantly, uninfected A549 cells from a separate flask were also trypsinized, stained, and sorted as "mock" population which served as a negative control.

## Single cell RNAseq cDNA library generation

Sorted cell samples were counted and checked for viability on a BD20 cell counter (BIO-RAD) before they were diluted to equivalent concentrations with an intended capture of 4000 cells/sample. Each individual sample was used to generate individually barcoded cDNA libraries using the 10x Chromium Single Cell 3' platform (Pleasanton, CA) following the manufacturer's protocol. The Chromium instrument separates single cells into Gel Bead Emulsions (GEMs) that facilitate the addition of cell-specific barcodes to all cDNAs generated during oligo-dT-primed reverse transcription. The experiment with Cal07 used V2 reagent and the experiment with Perth09 used V3 reagent (all steps followed the manufacturer's protocol).

## Illumina Library preparation and sequencing

Following ds-cDNA synthesis, individually barcoded libraries compatible with the Illumina chemistry were constructed. The libraries were sequenced on an Illumina NovaSeq 6000 using S4 flowcell for the experiment with Cal07 and S3 flowcell for the experiment with Perth09. Raw data can be found on the NCBI Gene Expression Omnibus under the GEO accession number GSE143167.

## Single cell RNAseq analysis

The three Cal07-associated 10x Chromium Single Cell v2 libraries (Infected, Bystander, and Mock) were demultiplexed and reference mapped using Cell Ranger Count (version 2.2) for alignment to a combined human+virus reference (human: hg38, version 1.2.0; Cal07: Genbank Accessions: CY121680-CY121687), and then combined using Cell Ranger Aggr. The three Perth09-associated 10x Chromium Single Cell v3 libraries (Infected, Bystander, and Mock) were processed and combined using Cell Ranger (version 3.1) for alignment to a combined human+virus reference (human: hg38, version 1.2.0; Perth09: Genbank Accessions: KJ609203.1- KJ609210.1). The resulting raw count matrix for each virus set was imported into an R pipeline using SimpleSingleCell [56], where it was filtered for empty droplets [57]. Cell cycle status was determined next by running the Cyclone tool in the scran R package [56]. Additional filtering was then performed on low feature cells (i.e. droplets removed if < 400 features/cell), low expressing features (i.e. features removed if < 4 cells/feature), and potential doublets (droplets removed if cell UMI counts > 2-fold of the median raw count number of host genes in cells of 3 libraries combined).

Overall cellular infection status (infected/uninfected) and individual virus gene presence/absence was determined by examining the distributions of percentage of viral counts within each cell on the log10 scale to magnify the differences at the low end. Thresholds for calling cells "not infected" overall and for individual viral gene presence/absence were set by calculating kernel density estimates on the distributions and finding the first local minima. For genes without clear bi-modal distributions (PB2 and PA of Cal07) the threshold was set to -1.5 (log10 scale), which was a consistent minimum in other low-expression viral genes and extremely close to the maximum value of PB2 and PA in uninfected cells within the Cal07 infected library. The filtered and annotated matrix was then imported into a Seurat pipeline for additional analysis and visualization [58], including normalization using the SCTransform method (all 3 libraries together for the combined tSNE but then each library separately for individual tSNE) [59], differential gene expression analysis using the MAST [25] and NBID tools [24], graph-based clustering of the cells [60], and PCA/tSNE dimensional reduction visualization.

Differential Gene Expression Analysis (DGE) for the Seurat clusters were performed by first sub-setting for 'Infected' status cells and then testing for each cluster versus all other clusters. The DGE gene list for each cluster consisted of genes with FDR < 0.01 in MAST test results generated from 10x Genomics Cell Ranger raw count matrix output. The DGE for missing individual virus genes were performed by first sub-setting the count matrices to contain only 'Infected' status cells and then using the individual virus gene status factors to test for cells with 'Present' versus 'Absent'. The DGE gene list for each viral segment was produced by intersecting genes with FDR < 0.01 in both MAST and NBID test results to minimize false positives.

All code used for single cell analysis, along with associated documentation, is available from: https://github.com/BROOKELAB/SingleCell

## Multivariate homogeneity of groups dispersions analysis

In the analyses, each host gene represents a variable/coordinate and thus each cell can be seen as a point in the multivariate space. Multivariate homogeneity of groups dispersion analysis [23] was performed on log transformed host gene expressions of the two group of cells, i.e. mock cells and infected cells, using the betadisper function in the R package (http://www.R-project.org/) vegan [61]. Distances between the points (i.e. cells) and their respective group centroid in the principal coordinates were then used to test homogeneity of variances and calculate the p-value. Note that similar analyses were performed on linear host gene expression values and results remain the same, i.e. overall host gene expressions are significantly more heterogenous than those in mock cells.

## DVG detection in viral stocks

Viral RNA was extracted from 140 ul of each viral stock tested using the QIAamp viral RNA kit (Qiagen) and eluted in 60 ul distilled water. For cDNA reactions, 3 ul of RNA was mixed with 1 ul (2 uM) MBTUni-12 primer (5-ACGCGTGATCAGCAAAAGCAGG-3), 1 ul (10 mM) dNTPs, and 8 ul distilled water. The mixture was incubated for 5 min at 65˚C and then placed on ice for 2 min. Subsequently, the mixture was removed from ice and the following were added: 1 ul SuperScript III (SSIII) RT (Invitrogen), 4 ul first-strand buffer, 1 ul DTT, and 1 ul RNase-in (Invitrogen). The reaction mixture was incubated at 45˚C for 50 min, followed by a 15 min incubation at 70˚C for inactivation. cDNA product (5 ul) was mixed with the following for PCR amplification: 2.5 ul (10 uM) MBTUni-12 primer, 2.5 ul (10 uM) MBTUni-13 primer (5-ACGCGTGATCAGTAGAAACAAGG-3), 0.5 ul Phusion polymerase (NEB), 10 ul high-fidelity buffer, 1 ul (10 mM) dNTPs, and 28.5 ul distilled water. The PCR protocol used was 98˚C (30 s) followed by 25 cycles of 98˚C (10 s), 57˚C (30 s), 72˚C (90 s), and a terminal extension of 72˚C (5 min). PCR products were run on 1% agarose gel and visualized on a BIO-RAD Gel Doc Universal Hood II Molecular Imager.

## Supporting information

**S1 Fig. Viral and host transcriptional similarity in cells from Cal07 infection.** tSNE dimensionality reduction plot showing the extent of overlap between 3 indicated cell populations from Cal07 experiment clustered based on transcriptional similarity.
(TIFF)

**S2 Fig. Determination of cutoff thresholds used to determine infection status. (A)** Histograms comparing distributions of total viral mRNA percentages in log10 scale across the three libraries (mock, bystander, and infected) for Cal07 experiment. Vertical dash line indicates cutoff threshold determined by calculating kernel density estimates on the distribution of infected library and finding the first local minima. **(B)** Cells from each library clustered as in S1 Fig are shown separately, with uninfected cells (below threshold) colored in cyan and infected cells (above threshold) colored in salmon. **(C)** Same figure as (A) for Perth09. **(D)** Same figure as (B) for Perth09.
(TIFF)

**S3 Fig. All pairwise Cal07 viral gene correlation plots.** Normalized per cell copy numbers for the indicated Cal07 genes plotted against each other. Data only show infected cells that are positive for all viral gene segments.
(TIFF)

**S4 Fig. All pairwise Perth09 viral gene correlation plots.** Normalized per cell copy numbers for the indicated Perth09 genes plotted against each other. Data only show infected cells that are positive for all viral gene segments.
(TIFF)

**S5 Fig. Determination of cutoff thresholds used to determine or presence/absence of individual viral gene segments. (A)** Histograms show the percentages of mRNA molecules derived from each Cal07 gene segment in log10 scale. Vertical dash lines indicate cutoff thresholds determined by calculating kernel density estimates on the distributions and finding the first local minima. For PB2 and PA that do not have clear bi-modal distributions, the threshold was set to -1.5 (log10 scale), which was a consistent minimum in other low-expression viral genes and extremely close to the maximum value of PB2 and PA in uninfected cells within the infected library. **(B)** Same figure as (A) for Perth09.
(TIFF)

**S6 Fig. Quantification of all differentially expressed host genes correlated with presence/absence of individual viral gene segments. (A)** The number of host transcripts for which expression levels significantly differ depending on whether the indicated Cal07 gene segment is present or not, according to both MAST and NBID (host genes that are differentially regulated by the expression status of more than one viral segment are included). **(B)** Same figure as (A) for Perth09.
(TIFF)

**S7 Fig. Comparison on DVGs content between validated viral stocks with low DVGs and viral stocks used in scRNAseq experiments.** PCR products following 8-segment whole-genome amplification from viral cDNAs of the pre-verified viral stocks (Cal07 LD and Perth09 LD have been shown to have minimum DVGs using NGS sequencing in articles published by the lab) and viral stocks used in scRNAseq experiments (Cal07 SC and Perth09 SC) are visualized on 1% agarose gel.
(TIFF)

**S1 Table. All host genes differentially expressed between Seurat clusters of Cal07 infected cells.**
(TXT)

**S2 Table. All host genes differentially expressed between Seurat clusters of Perth09 infected cells.**
(TXT)

**S3 Table. Combined DGE list based on Cal07 PB2 expression status within infected cells.**
(TXT)

**S4 Table. Combined DGE list based on Cal07 PB1 expression status within infected cells.**
(TXT)

**S5 Table. Combined DGE list based on Cal07 PA expression status within infected cells.**
(TXT)

**S6 Table. Combined DGE list based on Cal07 NP expression status within infected cells.**
(TXT)

**S7 Table. Combined DGE list based on Cal07 NA expression status within infected cells.**
(TXT)

**S8 Table. Combined DGE list based on Cal07 M expression status within infected cells.**
(TXT)

**S9 Table. Combined DGE list based on Cal07 NS expression status within infected cells.**
(TXT)

**S10 Table. Combined DGE list based on Perth09 PB2 expression status within infected cells.**
(TXT)

**S11 Table. Combined DGE list based on Perth09 PB1 expression status within infected cells.**
(TXT)

**S12 Table. Combined DGE list based on Perth09 PA expression status within infected cells.**
(TXT)

**S13 Table. Combined DGE list based on Perth09 HA expression status within infected cells.**
(TXT)

**S14 Table. Combined DGE list based on Perth09 NP expression status within infected cells.**
(TXT)

**S15 Table. Combined DGE list based on Perth09 NA expression status within infected cells.**
(TXT)

**S16 Table. Combined DGE list based on Perth09 NS expression status within infected cells.**
(TXT)

## Acknowledgments

We are grateful to Dr. Alvaro Hernandez and Ms. Chris Wright of the DNA Services Lab within the Roy J. Carver Biotechnology Center for expert assistance in experimental planning and the preparation and sequencing of single cell RNAseq libraries. We also would like to thank Dr. Chris Fields of HPCBio, also within the Roy J. Carver Biotechnology Center, for assistance in troubleshooting analysis.

## Author Contributions

**Conceptualization:** Christopher B. Brooke.

**Data curation:** Jiayi Sun, J. Cristobal Vera, Jenny Drnevich.

**Formal analysis:** Jiayi Sun, J. Cristobal Vera, Jenny Drnevich, Yen Ting Lin, Ruian Ke, Christopher B. Brooke.

**Funding acquisition:** Ruian Ke, Christopher B. Brooke.

**Investigation:** Jiayi Sun, J. Cristobal Vera, Jenny Drnevich, Yen Ting Lin, Ruian Ke, Christopher B. Brooke.

**Methodology:** J. Cristobal Vera, Jenny Drnevich, Yen Ting Lin, Ruian Ke, Christopher B. Brooke.

**Project administration:** Christopher B. Brooke.

**Resources:** Christopher B. Brooke.

**Software:** J. Cristobal Vera, Jenny Drnevich.

**Supervision:** Ruian Ke, Christopher B. Brooke.

**Validation:** Jiayi Sun, Jenny Drnevich, Christopher B. Brooke.

**Visualization:** Jiayi Sun, J. Cristobal Vera, Jenny Drnevich, Yen Ting Lin, Ruian Ke, Christopher B. Brooke.

**Writing – original draft:** Christopher B. Brooke.

**Writing – review & editing:** Jiayi Sun, Jenny Drnevich, Ruian Ke, Christopher B. Brooke.

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
