## [Decision Letter · Decision Letter 0]

21 Jan 2020

Dear Dr Brooke,

Thank you very much for submitting your manuscript "A common pattern of influenza A virus gene expression heterogeneity governs the innate antiviral response to infection" for consideration at PLOS Pathogens. As with all papers reviewed by the journal, your manuscript was reviewed by members of the editorial board and by several independent reviewers. In light of the reviews (below this email), we would like to invite the resubmission of a significantly-revised version that takes into account the reviewers' comments.

Thank you for submitting to PLoS Pathogens. I am sorry that this submission took so long.

All three reviewers very much appreciated the significant amount of data and work that went into this manuscript. They appreciated the ways in which this goes beyond other single cell studies of influenza virus.

All three reviewers raised points regarding the data related to NS, and suggested additional experiments that would solidify these findings. Given these concerns were common to all three, some sort of validation would be necessary here prior to publication. Several reasonable suggestions were made.

Reviewer 1 also had concerns about batch effects. This can be addressed by additional experimentation if feasible, or by caveat if not.

While some of the shortcomings mentioned by reviewer 3 regarding additional time points, cell lines, and strains would provide additional information, they would most likely not have a major impact on the conclusions reached from the current data. These are important points, however, and should be clearly discussed as limitations of the study in the discussion section.

We cannot make any decision about publication until we have seen the revised manuscript and your response to the reviewers' comments. Your revised manuscript is also likely to be sent to reviewers for further evaluation.

Sincerely,

Adam S. Lauring

Associate Editor

PLOS Pathogens

Andrew Pekosz

Section Editor

PLOS Pathogens

Kasturi Haldar

Editor-in-Chief

PLOS Pathogens

orcid.org/0000-0001-5065-158X

Michael Malim

Editor-in-Chief

PLOS Pathogens

orcid.org/0000-0002-7699-2064

Thank you for submitting to PLoS Pathogens. I am sorry that this submission took so long.

All three reviewers very much appreciated the significant amount of data and work that went into this manuscript. They appreciated the ways in which this goes beyond other single cell studies of influenza virus.

All three reviewers raised points regarding the data related to NS, and suggested additional experiments that would solidify these findings. Given these concerns were common to all three, some sort of validation would be necessary here prior to publication. Several reasonable suggestions were made.

Reviewer 1 also had concerns about batch effects. This can be addressed by additional experimentation if feasible, or by caveat if not.

While some of the shortcomings mentioned by reviewer 3 regarding additional time points, cell lines, and strains would provide additional information, they would most likely not have a major impact on the conclusions reached from the current data. These are important points, however, and should be clearly discussed as limitations of the study in the discussion section.

Reviewer's Responses to Questions

**Part I - Summary**

Reviewer #1: This manuscript from Vera et al. describes two single-cell sequencing datasets comprising infections by exemplar strains representing the circulating H1N1 and H3N2 viruses. The data are of considerable interest and utility to the field of influenza research, and the datasets are relatively high-quality and represent significantly rich resources for further work. Indeed, while the authors enrichment for influenza positive cells doubtless introduces some bias (an unavoidable fact in any perturbation), this permitted them probably the deepest and most comprehensive infectious dataset to date. The authors initial assessments of their datasets are also of interest. Of particular interest is their finding that the NS segment exhibits differential dynamics from the other seven flu segments, however I do believe this finding requires more support than is currently present in this manuscript. The authors also find a difference in the innate immune response to H3N2 and H1N1 influenzas and provide data suggesting differential reliance upon the NS segment (likely NS1) for suppression of an innate immune response.

With one considerable reservation, I believe this work represents a high-quality contribution to the field of influenza research and worthy of publication.

Reviewer #2: This manuscript by Vera et al examines the heterogeneity of antiviral responses to two strains of influenza virus using single cell RNAseq. They employ a clever flow cytometry pre sorting strategy to ensure that only 1 virion per cell. This allows them to examine a very large number of single infected cells, providing a degree of resolution which wouldn't be possible with conventional approaches. They discover significant cell-cell variability in virus replication levels, virus genes present, and host response. They further go on to directly compare two different strains H1N1 and H3N2 and uncover differences in their ability to induce interferon. This manuscript is well written, experiments are well done and the findings will be of great interest to readers in the field. However, there are some experiment that could significantly add to this study and extend it even further beyond what has been described previously. I also have some minor concerns/comments.

Reviewer #3: In this manuscript “A common pattern of influenza A virus gene expression heterogeneity governs the innate antiviral response to infection” by Vera et al., the authors examined viral and host transcriptomic changes during influenza A/California/07/2009 H1N1 and A/Perth/16/2009 H3N2 infection in A549 cells at the single cell level. Authors found different patterns of viral gene expression that resulted in multiple distinct host transcriptional responses to influenza viral infection. Moreover, the authors observed that differences in viral NS segment gene expression were responsible for shaping the host cell response to infection. Authors also observed that the pattern of host anti-viral gene transcriptional heterogeneity at the single cell level was different in cells infected with influenza H1N1 and H3N2 seasonal viruses.

Overall, this is an interesting manuscript for influenza virologists that provide a significant amount of information on viral and host transcriptomic responses, at the individual cell level, in A549 cells infected with seasonal H1N1 and H3N2 influenza virus. There are, however, several general concerns, mainly with the novelty and experimental approach, that reduce the overall enthusiasm for the manuscript.

**Part II – Major Issues: Key Experiments Required for Acceptance**

Reviewer #1: Major Comments

1) The authors use an orthogonal (ie sequenced separately albeit derived from same biological sample) bystander dataset to precondition their influenza thresholds. While I definitely appreciate the authors attempt to set an empirical threshold, the underlying approach they have taken has two considerable caveats that might seriously impact this body of work:

a. The first is the known influence of batch effects on single-cell datasets, an effect for which there has been considerable computational efforts to address. Indeed, the structure described by the tSNE in Figure 1B is likely as consistent with batch effects as it is with differential underlying biology. This would impact differential expression measurements between flu+ and flu- datasets, but, more critically in my mind, could lead to incorrectly derived thresholds for presence/absence of an influenza gene.

b. The second is that any lysis (or other means or RNA release) that occurs post-sorting, either in the microfluidics or even while the cells are sitting after sorting, would not be captured in the bystander dataset. This would also lead to incorrectly derived thresholds.

Truthfully, I am not sure there is a way to “fix” this problem without repeating these experiments. To be clear, this is a problem which is present in many similar, published, datasets. I do not think it is a strong enough issue to warrant repeating these experiments to include some truly internal error correction control, however I do believe the authors could change the text to be much more upfront about these considerations. As written, my concern is the authors claim this as a much stronger error-correction method than I believe is warranted and so these issues might propagate in future studies.

2) Somewhat following from my first concern, I am a bit troubled that some of the NS segment dynamics could be explained by an “incorrect” threshold. In Figure 5D there appear to be two populations, “NS-low” and “NS-high”, that each exhibit parallel behavior with respect to their relationship with the rest of the flu segments. If NS-low is really NS null, and the increasing flu counts is more a function of capture efficiency (or reverse-transcription efficiency) than any underlying biological difference, then it is possible that the authors are merely describing segment absence, a phenomenon already described in great detail by this group. That this problem might be restricted to, or more severe in, the NS segment would be somewhat expected, as it would likely track with relative expression (ie more expression leading the greater extracellular contamination), and NS and M are the most highly expressed segments. The absence of this effect in M could then be explained by the deliberate sorting on M2. I think if the authors want to make this point, which is of interest and is aberrant with respect to other measurements, then an orthogonal means of measurement must be employed. I think the most straightforward would be to measure either NEP or NS1 and some other flu protein by flow cytometry and see if this population can be detected. Alternatively, if there are no reasonable antibodies against NS1/NEP (there are for WSN, however I admit ignorance for the cross-reactivity with circulating H1N1 and H3N2), the authors could try and use reporter strains (bearing fluorescent proteins) in the NS segment instead. Admittedly the absence of the phenotype in those populations could also be due to perturbations caused by generation of a reporter virus, but the authors should at least make the attempt. If neither of these experiments is possible, it would be ideal to state as such in the text and the authors MUST at least address this reasonable alternative hypothesis in a manner which is clear and upfront and obvious to the reader.

Reviewer #2: The finding that Slfn5 is inversely correlated with NS is extremely interesting. Can the authors perform follow up analysis using NS1 deficient viruses to show they get the predicted phenotype in bulk A549 cells. Additionally can they use Slfn5 ko to demonstrate a consequence for this observation?

Reviewer #3: 1) One of the concerns with this manuscript is the novelty. As indicated by the authors in several parts of the document, similar studies to those described in this manuscript have been previously conducted and reported in the literature by different research groups (e.g., Extreme heterogeneity of influenza virus infection in single cells; Single-cell analysis and stochastic modelling unveil large cell-to-cell variability in influenza A virus infection; Innate immune response to influenza virus at single-cell resolution in human epithelial cells revealed paracrine induction of interferon lambda 1; among others). Thus, it is unclear the new findings in this manuscript. At least the authors should clearly outline how the studies in this manuscript differ from those in previously published literature.

2) Another concern is that the results are based on the infection of a single cell type (A549) at a single time post-infection (16 hours). It is possible, and most likely, that different results will be obtained using a different cell line and different times post-infection.

3) Experiments to evaluate changes in viral and/or host transcriptomic responses at the single cell level from natural influenza human infections would be more relevant than the use of A549 cells. Alternatively, the use of more relevant primary airway culture systems rather than a cancer cell line would increase the relevance of this study. Moreover, looking at different times post-infection, rather than a single time point, to assess the dynamic changes in viral and host transcriptomic responses during viral infection will probably increase the significance of this study.

4) Another concern is the lack of synchronization of viral infections. It is possible that both viral and host transcriptomic differences are due to differences in the replication step of the virus in different cells. Synchronizing viral infections would allow to rule out this possibility.

5) The authors indicate that changes in viral and host transcriptomics were obtained from single virion-infected cells based on the use of low multiplicity of infection (MOI; 0.01). However, this has not been clearly shown. Likewise, the authors indicate that viral stocks lack defective interfering particles (DIPs) since they were generated under experimental conditions that minimize the presence of DIPs (e.g. low MOI). However, the presence/absence of DIPs in viral stocks have not been clearly evaluated and could be responsible, at least in part, of the differences in the viral and/or host transcriptomic responses.

6) To clearly demonstrate the role of the viral NS1 in the differences of host transcriptomic responses, the authors should consider the possibility of comparing results from wild-type (WT) and NS1-deficient H1N1 and/or H3N2 virus infected cells.

7) The authors observed differences in both viral and host transcriptomic responses between H1N1 and H3N2 seasonal viruses. It is possible, and most likely, that differences in viral replication/transcription dynamics of these two viruses are responsible of these disparities.

8) Likewise, the authors suggest that differences in antiviral transcriptomic responses between H1N1 and H3N2 seasonal viruses could be attributed to differences in the NS1 proteins. However, it is unclear if the authors have compared the ability of the NS1 proteins from H1N1 and H3N2 seasonal viruses to inhibit innate immune responses and host gene expression; and whether or not these differences are responsible for those seen in the host transcriptomic responses in infected cells. Moreover, to clearly demonstrate that the NS1 protein and no other viral proteins are responsible for the differences in host transcriptomic responses between H1N1 and H3N2 seasonal viruses, the authors should generate recombinant viruses with the same genetic backbone that only differ in the viral NS1 protein.

9) As indicated in the manuscript, influenza PA-X has been described to inhibit host gene expression during influenza viral infection. It is unclear if the differences observed between the two seasonal influenza H1N1 and H3N2 viruses could be explained based on the ability of their PA-X proteins to induce cellular host shut off.

10) Some of the figures were difficult to assess due to the low quality, mainly Figures 2, 3, 6, and 7.

**Part III – Minor Issues: Editorial and Data Presentation Modifications**

Reviewer #1: 1) Line 67. The authors might have a comment to say about other datasets, but I find it relatively vague. I would appreciate either directly stating the “experimental design choices that complicated analysis” to inform the rest of the field, or leaving this sentence out entirely. This work stands on its own, but if the authors wish to make a comment to directly improve the field they should probably be explicit in their recommendations.

2) Line 90. This is an incorrect calculation as known from the senior author’s own work. While calculations of SIP frequency vary, the lower bounds are still around 10:1 SIP-non-SIP particles. If the authors are using an MOI of 0.01, the effective particle dose would then be at least 0.1, which emphatically would not lead to greater than 99.99% of infected cells experiencing only one virion. The MOI the authors use is certainly low enough for their experimental design and I would be perfectly happy with them removing the incorrect calculation from the text.

3) A bit on the same theme as the major comments above I am afraid. If the thresholds the authors set are biologically meaningful, the absence of the polymerase segment would be predicted to lead to less influenza transcripts observed, do the authors note this? Similarly, we might expect absence of polymerase segments to drive similar (not necessarily identical) transcriptional programs in the host, can you comment on this?

4) Lines 111-113 are a bit strong, there is probably still some non-zero rate of coinfection (particularly given the incorrect assumptions of particle ratios discussed above), and whether a 1 hour infection window is “tight” probably depends on who you ask. Slightly more equivocal language would be appreciated.

5) Line 229. This is written in a misleading fashion and needs to be fixed. Russell et al. did describe a statistical association with NS absence and induction of interferon, even though it was non-deterministic. The way I read this sentence is that there was no association found (given that you are claiming your results “echo” this previous finding). This does not detract from the authors’ observations, as they performed their experiment with a different strain under different conditions, so if they could simply fix the text to be a bit less misleading that would be perfect.

6) Lines 240-241. The use of R2 here, and the language used, doesn’t appear to reflect what the authors observe. It looks as if there are two NS populations (as stated above, NS-low and NS-high) both of which appear to exhibit positive correlations with other influenza segments. This is stated a bit more clearly in lines 257-259, but I feel would lead to considerable misunderstanding as written. By conflating two subpopulations into a single R2 measurement there is the dubious conclusion that there is no positive correlation. If the authors could clearly state as such, and even calculate the R2 for each population (even some arbitrary threshold would be fine, say 102 copies of NS1) they would likely much better capture the dynamics of their observations.

7) Lines 347-361. While not the exact strain used here, a previous study (Hayman, A. et al. Variation in the ability of human influenza A viruses to induce and inhibit the

IFN-β pathway. Virology 347, 52–64 (2006).) Found a similar increased interferon induction by a few circulating H3N2 strains. I think probably worth citing and a quick mention.

Reviewer #2: -What percent of cells that are HA-M2- at 16 hrs become either HA or M2+ at later time points? This might help understand some of the minor limitations of the flow based pre screen and would help add to the body of literature on the kinetics of IAV protein and mRNA expression heterogeneity at single cell levels.

-What is the rationale for only using top 10 host genes for Fig 2 D-E? Shouldn’t the top 100 be used given the range of virus levels?

-Can the authors provide information one the transcriptional responses to the extreme ends of the virus replication levels (1-90%)? Could this be a stronger driver than presence/absence of specific influenza genes?

-Another potential explanation for lack of polymerase still having surface HA and M2 is production through primary transcription alone, which could be mentioned.

-Can the authors speculate as to why PB2 is the most frequently missing segment? And what consequences this might have for reassortment etc

Reviewer #3: None

PLOS authors have the option to publish the peer review history of their article (what does this mean?). If published, this will include your full peer review and any attached files.

Reviewer #1: No

Reviewer #2: No

Reviewer #3: No
---

## [Decision Letter · Decision Letter 1]

1 Jun 2020

Dear Dr Brooke,

We are pleased to inform you that your manuscript 'Single cell heterogeneity in influenza A virus gene expression shapes the innate antiviral response to infection' has been provisionally accepted for publication in PLOS Pathogens.

Best regards,

Adam S. Lauring

Section Editor

PLOS Pathogens

Andrew Pekosz

Section Editor

PLOS Pathogens

Kasturi Haldar

Editor-in-Chief

PLOS Pathogens

orcid.org/0000-0001-5065-158X

Michael Malim

Editor-in-Chief

PLOS Pathogens

orcid.org/0000-0002-7699-2064

This is a nice piece of work. Thank you for submitting to PLOS Pathogens and for your efforts to address the reviewers' comments.

Reviewer Comments (if any, and for reference):

Reviewer's Responses to Questions

Part I - Summary

Reviewer #1: The authors have appropriately addressed my remaining misgivings. i see no reason why the publication of this manuscript should be delayed further, as it represents a solid dataset and reasonable analysis that is of general interest to the audience of PLoS Pathogens.

Reviewer #2: In this revised manuscript by Sun and colleagues address the key critiques of all three reviewers, resulting in a significantly strengthened manuscript. They have added substantial new analysis and clarified several key points. This is a very rigorous assessment of the replication heterogeneity after influenza infections and uses several clever experimental and computational approaches. They are able to overcome several key caveats of other studies allowing for a highly refined assessment of infection heterogeneity. Overall the manuscript is well written and the experiments are well designed and executed. The biological findings and approaches presented will be of significant interest to readers in the field.

Part II – Major Issues: Key Experiments Required for Acceptance

Please use this section to detail the key new experiments or modifications of existing experiments that should be 

absolutely

 required to validate study conclusions.

Reviewer #1: (No Response)

Reviewer #2: (No Response)

Part III – Minor Issues: Editorial and Data Presentation Modifications

Reviewer #1: (No Response)

Reviewer #2: (No Response)

PLOS authors have the option to publish the peer review history of their article (what does this mean?). If published, this will include your full peer review and any attached files.

Do you want your identity to be public for this peer review?

 For information about this choice, including consent withdrawal, please see our Privacy Policy.

Reviewer #1: No

Reviewer #2: No

---

## [Editor Report · Acceptance letter]

22 Jun 2020

Dear Dr Brooke,

We are delighted to inform you that your manuscript, "Single cell heterogeneity in influenza A virus gene expression shapes the innate antiviral response to infection," has been formally accepted for publication in PLOS Pathogens.

Best regards,

Kasturi Haldar

Editor-in-Chief

PLOS Pathogens

orcid.org/0000-0001-5065-158X

Michael Malim

Editor-in-Chief

PLOS Pathogens

orcid.org/0000-0002-7699-2064